

# Analysis of RecA-independent recombination events between short direct repeats related to a genomic island and to a plasmid in *Escherichia coli* K12

María F. Azpiroz[*] and Magela Laviña[*]

Fisiología y Genética Bacterianas, Facultad de Ciencias, Universidad de la República, Montevideo, Uruguay
[*] These authors contributed equally to this work.

## ABSTRACT

RecA-independent recombination events between short direct repeats, leading to deletion of the intervening sequences, were found to occur in two genetic models in the *Escherichia coli* K12 background. The first model was a small *E. coli* genomic island which had been shown to be mobile in its strain of origin and, when cloned, also in the *E. coli* K12 context. However, it did not encode a site-specific recombinase as mobile genomic islands usually do. It was then deduced that the host cells should provide the recombination function. This latter was searched for by means of a PCR approach to detect the island excision in *E. coli* K12 mutants affected in a number of recombination functions, including the 16 *E. coli* K12 site-specific recombinases, the RecET system, and multiple proteins that participate in the RecA-dependent pathways of homologous recombination. None of these appeared to be involved in the island excision. The second model, analyzed in a RecA deficient context, was a plasmid construction containing a short direct repeat proceeding from *Saccharomyces cerevisiae*, which flanked the *cat* gene. The excision of this gene by recombination of the DNA repeats was confirmed by PCR and through the detection, recovery and characterization of the plasmid deleted form. In sum, we present new evidence on the occurrence of RecA-independent recombination events in *E. coli* K12. Although the mechanism underlying these processes is still unknown, their existence suggests that RecA-independent recombination may confer mobility to other genetic elements, thus contributing to genome plasticity.

## INTRODUCTION

Bacterial genomes contain short repeated DNA sequences, which are prone to recombine and generate genetic rearrangements. Some of them are associated with mobile genetic elements such as prophages and genomic islands, whose insertion generates short, direct repeats on either side of the element. These flanking repeats, known as attachment sites, are usually the target of dedicated site-specific recombinases that excise these elements so that they remain as independent covalently closed circular molecules. In the inverse reaction, the same recombinases integrate the circular phage or island into the corresponding

Corresponding author
María F. Azpiroz,
fernanda@fcien.edu.uy

attachment site in the genome (*Hallet & Sherratt, 1997*; *Hochhut et al., 2006*; *Lin et al., 2008*; *Murphy & Boyd, 2008*; *Juhas et al., 2009*).

The present work continues previous studies on the mobility of a genomic island, called H47 (H47 GI), contained in the chromosome of an *Escherichia coli* strain (H47). It is a small 13 kb-genetic element devoted to the production of the antibacterial peptide microcin H47 (*Laviña, Gaggero & Moreno, 1990*; *Poey, Azpiroz & Laviña, 2006*). The H47 GI was found to be an unstable mobile element, able to excise from the chromosome by recombination between its attachment sites (*attL* and *attR*) in a process generating two products: the chromosome without the island and the island as an independent covalently closed circular molecule (Fig. 1A) (*Azpiroz, Bascuas & Laviña, 2011*). A DNA segment containing the island had been cloned into several multi-copy plasmid vectors and introduced into *E. coli* K12 cells. In this foreign background, the H47 GI retained its mobility as in the original H47 strain. In addition, it was found that the excised island was able to integrate into another plasmid provided that it contained the corresponding attachment site. These findings were attained by means of PCR experiments, which allowed the detection of both recombinant sequences generated from each type of recombination event, excision and integration (*Azpiroz, Bascuas & Laviña, 2011*). The H47 GI is flanked by two extensive and imperfect direct repeats, *attL* and *attR*, which share four main regions of homology (Fig. 1B). In the case of excision, *attL* and *attR* recombine and the two resulting products were detected by the appearance of amplicons that included the recombined attachment sites: *attC* in the deleted replicon and *attI* in the excised H47 GI (Fig. 1C). Amplicons containing the *attC* site were a mix of four sequences corresponding to recombination at the four main regions of homology between *attL* and *attR*. This recombination pattern remained unchanged when the analysis was carried out in RecA-deficient *E. coli* K12 cells, indicating that the H47 GI mobility is a process independent from the homologous recombination pathway (Fig. 1C and S1) (*Azpiroz, Bascuas & Laviña, 2011*).

Many reports refer that genomic islands usually undergo mobility events, and that these are catalyzed by site-specific recombinases encoded by the islands themselves, a feature that is shared with temperate phages (*Hochhut et al., 2006*; *Juhas et al., 2009*). However, the H47 GI does not encode any recombinase: when most of its DNA was deleted, recombination between its attachment sites remained unchanged (*Azpiroz, Bascuas & Laviña, 2011*). Therefore, the genomic background of *E. coli* H47 and of *E. coli* K12 should provide the recombination function responsible for the H47 GI mobility. The island would thus employ a sort of parasitic strategy and, given the characteristics of the rearrangements observed, a site-specific recombination process was expected to accomplish the genetic exchange.

In this work, we searched for the function underlying the H47 GI mobility, working with the cloned island in *E. coli* K12 cells. The H47 GI excision was PCR-assayed in a broad collection of mutants affected in the site-specific and homologous recombination pathways encoded by *E. coli* K12. None of these functions affected the island mobility, indicating that the process under study could be included into the poorly-known group of RecA-independent recombination events (*Bzymek & Lovett, 2001*). However, we could not succeed to obtain a clone carrying the deleted form with the *attC* site. To gain further insight into this type of process, a plasmid construction containing the repeated FRT (Flp

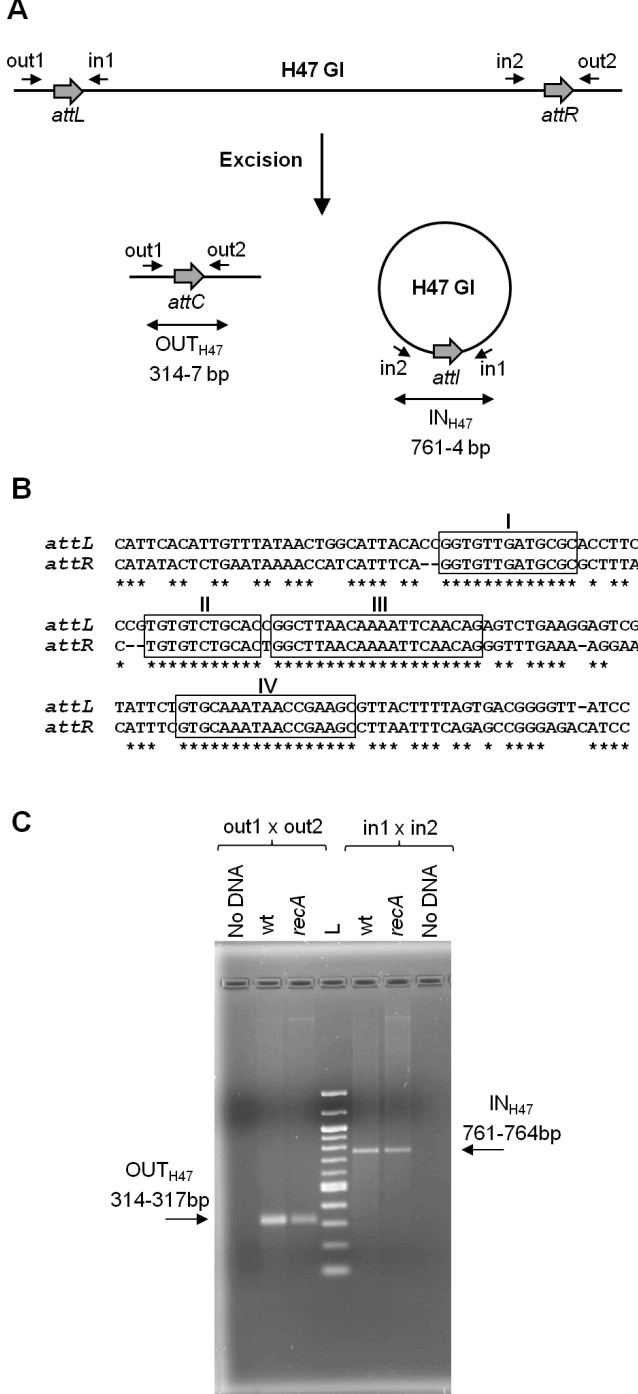

**Figure 1 Excision of the H47 genomic island.** (A) PCR approach performed to detect the excision of the H47 GI. Attachment sites are shown with grey thick arrows and primers with small arrows. Products of excision were detected by the appearance of amplicons obtained with the indicated primer pairs: amplicon OUT$_{H47}$, containing the *attC* site, and amplicon IN$_{H47}$, containing the recombined attachment site *attI* in the excised H47 GI. (B) Alignment of the *attL* and *attR* sites. Identical nucleotides are indicated with asterisks. The four main regions of homology are boxed and named with roman numbers. (C) Detection of the H47 GI excision using as template pEX2000 propagated in *E. coli* K12 BZB1011 (wt), and in its derivative mutant *recA*. L: 100 pb DNA Ladder (New England Biolabs, Ipswich, MA, USA).

recognition target) site from *Saccharomyces cerevisiae* was similarly analyzed in the context of *E. coli* K12. In this model, RecA-independent deletion of the intervening sequence between the repeats was detected *in vitro* by PCR and *in vivo* by the appearance of the deleted form in sufficient amounts to be recovered from plasmid DNA preparations.

## MATERIALS & METHODS

### Bacterial strains, plasmids, and growth conditions

*E. coli* K12 BZB1011 is a spontaneous nalidixic acid resistant derivative of W3110, a strain whose genome has been sequenced (GenBank: AP_009048). Most strains used are BZB1011 derivatives, carrying deletions proceeding from selected mutants of the Keio Collection (*Baba et al., 2006*). In these latter, gene deletions with replacement by a kanamycin resistance (*kan*) cassette were introduced into BZB1011 by P1 transduction, selecting the kanamycin resistant clones (Table S1). When required, double deletion mutants were constructed in two steps: the kanamycin cassette of the first mutated gene was eliminated as indicated previously (*Datsenko & Wanner, 2000*) and the cured mutant was then transduced with a P1 grown on the second deletion mutant, selecting the kanamycin resistant clones. Multiple deletion mutants were constructed with further rounds of the same procedure. *E. coli* K12 JC8679 is a *recB recC sbcA* mutant (Coli Genetic Stock Center strain 6490). Two mutant derivatives of BW25113, the parent *E. coli* K12 strain for the Keio Collection, were used: BW25113 *recA56* and BW25113 *topB recA56*, this latter bearing the Δ*topB761::kan* mutation from the Keio Collection. The *recA56* allele was introduced by transducing BW25113 and BW25113 *topB* (JW1752) with P1 grown on a *srl::* Tn*10 recA56* strain from our laboratory collection, selecting the tetracycline resistant clones and then assaying the *recA* defficiency by testing their UV sensitivity.

Plasmid pEX2000 is a pBR322 derivative that carries the H47 GI included in a 16,823 bp-chromosomal DNA segment from strain H47 (GenBank: AJ_009631). Plasmid pΔint1 carries the same DNA segment but deleted for most of the H47 GI, keeping the attachment sites and the external chromosomal adjacent sequences (*Azpiroz, Bascuas & Laviña, 2011*). pUY-FRT was constructed by cloning a 1,182 bp HindIII fragment from pKD3 (GenBank: AY_048742) into the pUC13 vector. The insert contains the chloramphenicol resistance gene (*cat*) flanked by the FRT sites from *S. cereviseae*. Since pKD3 requires the *pir* gene product to be provided in trans for its replication (*Datsenko & Wanner, 2000*), pUY-FRT was constructed to avoid this requirement.

Bacteria were grown in LB medium at 37 °C. Antibiotics were added at the following final concentrations: kanamycin, 30 µg/ml; ampicillin, 50 µg/ml, chloramphenicol, 60 µg/ml, and tetracycline, 20 µg/ml.

### Plasmid purification, PCR assays and DNA sequencing

Plasmid DNA was extracted with "QIAprep Spin Miniprep Kit" (Qiagen, Hilden, Germany). DNA bands were extracted from gels using "QIAEXII Gel extraction kit" (Qiagen, Hilden, Germany). For PCR reactions, primers and conditions employed are explained in Table 1. For sequencing purposes, PCR products were extracted from gels as explained. Both strands of the OUT and IN amplicons from the pUY-FRT model were

Azpiroz and Laviña (2017), *PeerJ*, DOI 10.7717/peerj.3293

**Table 1** Conditions of PCR reactions.[a]

| Model | Primer name[b] | Primer sequence 5'5' | Annealing temperature | DNA template (ng) | Product name and expected size (bp) |
|---|---|---|---|---|---|
| H47 GI excision from pEX2000 | out1 | CCGTTCATTTTCCTGCTGACCC | 58° | 200–400 | $OUT_{H47}$ (314–317) |
| | out2 | TCTGTTGCCCGTTGATGTTTCCT | | | |
| | in1 | GTTTGTAGGAGCTTTCTTTTTTG | 53° | 200-400 | $IN_{H47}$ (761–764) |
| | in2 | CGCTGATGACTGTTTTTATGTTG | | | |
| *cat* excision from pUY-FRT | f | GTTGTAAAACGACGGCCAGT | 58° | 200–300 | $OUT_{FRT}$ (381) |
| | r | CACAGGAAACAGCTATGACC | | | |
| | II-in1 | AAGGCGACAAGGTGCTGATG | 58° | 200-300 | $IN_{FRT}$ (335) |
| | II-in2 | GGAACCTCTTACGTGCCGAT | | | |

**Notes.**

[a]PCR amplifications were performed using U-Taq DNA polymerase (SBS Genetech) in a total volume of 30 μl. Reaction mixes contained 1× buffer, 200 μM of each deoxynucleotide triphosphate, 500 nM of each primer, 1.25 U of DNA polymerase and template DNA. Conditions for amplification were: 2 min at 94 °C, 30 cycles of incubation at 94 °C for 30 s, annealing temperature for 30 sec, 72 °C for 30 s, and a final extension step at 72 °C for 2 min.

[b]Primers used to assay the H47 GI model were presented in *Azpiroz, Bascuas & Laviña (2011)*. F and R are M13 forward and reverse primers. II-in1 and II-in2 were designed in this work.

sequenced using the primers employed for their amplification. DNA sequencing was performed at the "Molecular Biology Unit" of the Pasteur Institute of Montevideo.

## RESULTS

### Excision of the H47 genomic island

Recombination underlying the H47 GI mobility was studied through the detection of the island excision from a multi-copy recombinant plasmid, pEX2000, in the context of *E. coli* K12 cells. Plasmid pEX2000 carries a DNA segment from the *E. coli* H47 chromosome containing the H47 GI flanked by its direct repeats and by adjacent chromosomal sequences on both sides. The direct repeats, *attL* (148 bp) and *attR* (144 bp), are imperfect and share four main segments of homology of 13, 11, 20 and 17 bp (Fig. 1B). Recombination between them leads to the excision of the H47 GI, which is assessed by PCR using primers "out" in order to detect the occurrence of the empty *attC* site (amplicon $OUT_{H47}$) (Fig. 1A). The assays were performed in the context of the *E. coli* K12 strain BZB1011 and in a set of BZB1011 derivative mutants affected in recombination functions (Table S1). The purpose was to identify the mutant/s where excision did not occur, thus revealing the gene/s involved in the recombination process. Plasmid pEX2000 was introduced into each of these strains and then was extracted to be used as template in PCR reactions. The sequence of the OUT amplicons corresponding to the mutant strains was compared with that from the wild-type BZB1011 strain, i.e., with a mix of four overlapping sequences with the predominance of that corresponding to recombination in site III (Fig. S1).

Considering that genomic islands usually employ site-specific recombinases to mediate their mobility, the search first concentrated in mutant strains deficient for such enzymes. Although the H47 GI attachment sites did not reveal significant homology with any known target for a site-specific recombinase, we did not discard this possibility since it has been reported that some of these enzymes are able to work although with low efficiency—on secondary sites that differ in their sequence from the primary one (*Menard & Grossman, 2013*). Therefore, the excision of the H47 GI was analyzed in mutants deficient for each of the 16 site-specific recombinases encoded by the *E. coli* K12 chromosome, most of them related to prophages (*intF, intD, ybcK, intE, pin, intR, pinR, pinQ, intQ, intS, intA, xerD, xerC, intB, fimB* and *fimE*). Mainly based on homologies among these enzymes, double and multiple mutants were constructed and included in the assays (Table S1). In all cases the $OUT_{H47}$ amplicon appeared and always revealed the same pattern of overlapping sequences as that proceeding from the wild-type BZB1011 strain (Fig. S1).

Given the previous results, the involvement of site-specific recombination in the H47 GI mobility was discarded. An important fact was that recombination always took place in regions of total homology between the attachment sites, indicating that this condition would be necessary. Therefore, we examined the possible involvement of several genes that participate in homologous recombination.

First, the RecA-independent RecET homologous recombination pathway, encoded by the defective Rac prophage present in *E. coli* K12, was analyzed (*Kolodner, Hall & Luisi-DeLuca, 1994*). It was considered that the *recET* genes might have a basal level of

expression, as has been proposed for *E. coli* K12 crytic prophage genes (*Wang et al., 2010*). The excision of the H47 GI was assayed in BZB1011 mutant derivatives *recE* (deficient for exonuclease VIII) and *recT* (deficient for the recombinase function). None of these contexts affected the recombination pattern. The same result appeared using the *recB recC sbcA* strain JC8679, a genetic background where the RecET pathway is known to be induced (*Kolodner, Hall & Luisi-DeLuca, 1994*).

It was also considered that recombination under study, although RecA-independent, could share some functions with the homologous recombination pathways. Following this idea, the excision of the H47 GI was surveyed in a set of mutants defective for proteins that participate in homologous recombination (*recB, recC, recD, recF, recO, recR, recN, recJ, recQ, recG, sbcB, sbcC, sbcD, uvrD, ruvA, ruvB, ruvC, radA, recX, exoX, rarA, seqA* and *helD*). The double mutant *recT recA*, deficient for all known homologous recombination pathways, was constructed and also included in the assays. The result was that in neither mutant the excision of the island was detectably affected: the OUT$_{H47}$ product always appeared as a weak band in gels and its sequence exhibited the same pattern as that of the amplicon from the wild-type strain (Fig. S1). Therefore, none of the functions analyzed related to homologous recombination appeared to be involved in the H47 GI mobility.

### *In vivo* search for the H47 GI excision

Although *in vitro* PCR experiments consistently supported the existence of recA-independent recombination events between the *attL* and *attR* sites flanking the H47 GI, this phenomenon still awaited an *in vivo* confirmation. This would imply the detection and characterization of at least a clone carrying the replicative recombination product, i.e., the deleted molecule. Obviously, the *in vivo* detection of the other putative product, the excised circular form, would be much less likely because of the non-replicative condition of this molecule. It should be noted that the model of the H47 genomic island was rather complex since it encodes the production of an antibiotic activity, which would exert a deleterious effect upon cells that had lost the island in a recombination event. For this reason, we investigated this phenomenon not only in strains carrying the entire island (pEX2000) but also in strains with a plasmid derivative in which most of the intervening sequence between the *att* sites had been deleted (*p*Δint1).

After being propagated in *E. coli* K12 cells (BZB1011 *recT recA*), plasmids were extracted, and important amounts of DNA (1–2 μg) were run in gels in an attempt to perceive new bands that could correspond to the recombination products. We did not succeed to see anything other than the plasmids' original forms.

It has been described that *E. coli* mutants affected in the *topB* gene, encoding topoisomerase III, have an increased frequency of spontaneous deletions occurring at short direct repeats (*Whoriskey, Schofield & Miller, 1991*; *Schofield et al., 1992*; *Uematsu, Eda & Yamamoto, 1997*). Therefore, we repeated the assays in the context of strains carrying a *topB* deficient allele. BZB1011 *topB* was constructed, but its *recA* derivative resulted nonviable. Considering that the *topB* context would be affected in DNA supercoiling, the nalidixic acid resistant strain BZB1011 might not be suitable for this study since it most probably contains a mutant DNA gyrase. For this reason, the *gyr* wild type BW25113 context was

employed. BW25113 *recA* and BW25113*topB recA* were constructed and transformed with pEX2000 and pΔint1. In all these strains, although recombination between the *att* H47 sites was PCR-confirmed, no extra-bands appeared when plasmids DNA was analyzed in gels.

## H47 GI-related *attC* sequence survey in data banks

Since no product of the H47 GI excision could be detected *in vivo*, a clue in this sense was searched for in data banks. This type of survey had been done before, mainly focusing on the presence of the H47 GI in other strains besides *E. coli* H47 (*Azpiroz, Bascuas & Laviña, 2011*). Now, we looked for the four types of *attC* sequences (I–IV), corresponding to recombination events in each of the four segments of identity between the *attL* and *attR* sites. The search revealed that these sequences are widespread among pathogenic *E. coli* strains and also appear in two strains of *Salmonella enterica*, always being present in the chromosome. Although these sequences exhibited a certain degree of variability, several *attC* sites could be recognized and distinguished from *attL* and *attR*. They were further confirmed as *attC* by analyzing their adjacent sequences (about 500 nucleotides on each side), which should correspond to those surrounding the H47 GI. Most of them were $attC_{III}$, resulting from recombination at the most extensive site of homology between the direct repeats. Among these, six identical matches were found (*E. coli* strains Sanji, SEC470, RS76, PCN033, Santai O157:H16, and *S. enterica* Heidelberg str. SL476). There was a match with $attC_{II}$ (*E. coli* ETEC H10407) and another single match with $attC_{IV}$ (*E. coli* FHI23). No sequence of the $attC_I$ type appeared. Therefore, these findings strongly supported the idea that the H47 GI is indeed a mobile element due to recombination between its attachment sites.

## RecA-independent recombination in the model of plasmid pUY-FRT

In view that our efforts to detect *in vivo* the H47 GI excision were unsuccessful, the studies were extended to a simpler genetic model. It was a multi-copy plasmid, pUY-FRT, carrying the *cat* gene (for chloramphenicol resistance) flanked by a perfect 46 bp-direct repeat. The repeated sequences proceeded from *S. cereviseae* and contained the FRT target for the site-specific recombinase Flp. Recombination between the repeats would lead to the deletion of the intervening *cat*-containing DNA segment. The assays were carried out in the absence of the Flp enzyme in *E. coli* K12 cells. Specifically, plasmid pUY-FRT was propagated and then extracted from BZB1011 *recT recA*, BW25113 *recA* and BW25113 *topB recA*.

The experiments began by transforming with pUY-FRT the three *E. coli* K12 mutant backgrounds and chloramphenicol resistant transformants were selected so as to ensure that the assays started with clones carrying the original non-deleted form of the plasmid. After isolation in the same medium, clones were grown in liquid LB Ap, which was supposed to be a permissive condition for the propagation of the deleted plasmid.

First, plasmid DNA was used as template in PCR reactions devoted to detect the excision of the DNA segment between the repeats, i.e., the *cat* gene. As with the H47 GI, two types of reactions were carried out to detect the two possible products: an $OUT_{FRT}$ amplicon of 381

bp in the deleted plasmid, and an $IN_{FRT}$ amplicon of 335 bp in the excised circular form. In both types of reactions, amplicons of the expected size were produced and appeared in the three contexts assayed. The DNA sequence of these PCR products exhibited the expected recombination pattern, i.e., a single repeat flanked by recombined sequences.

For *in vivo* analysis, ca. 500 ng of plasmid DNA were loaded in gels. This time, a new small band was barely seen in plasmid preparations proceeding from the $topB^+$ backgrounds, being more evident in the BW25113 *recA* context, and this same band appeared clearly stronger when the plasmid came from the *topB* deficient cells. We presumed that it could correspond to the deleted plasmid molecule, which would be called pUY-FRTΔ (Fig. 2A). Then, this smaller band, proceeding from BW25113 *recA* and BW25113 *topB recA*, was extracted from gels and used to transform BW25113 *recA* with selection for ampicillin-resistance. In each case, a few clones grew and, when assayed in the presence of chloramphenicol, all proved to be sensitive. Plasmid DNA was extracted from a clone of each type of transformant and was then digested with EcoRI and HindIII. Both plasmids were found to be identical in their size and restriction profile, which corresponded to those of a deleted plasmid derived from pUY-FRT by recombination between its direct repeats (Fig. 2B). DNA sequencing confirmed this structure: the sequence was read through the recombined repeat and beyond it more than 300 bp on each side.

Finally, it should be mentioned that no trace of the excised DNA could be seen.

## DISCUSSION

The analysis of the mobility of an *E. coli* small genomic island led us to study RecA-independent recombination phenomena. Specifically, the H47 GI presented the peculiarity of being able to excise from its site of insertion although it did not code for a cognate integrase. Moreover, its mobility was not affected in a RecA deficient background (*Azpiroz, Bascuas & Laviña, 2011*). It should be mentioned that a few other genetic elements have been described to be mobile and to lack a recombinase gene (*Santiviago et al., 2010*; *Palmieri, Mingoia & Varaldo, 2013*).

The fact that the H47 GI mobility could be analyzed in the *E. coli* K12 genetic context appeared as an excellent opportunity to identify the host function involved in the genetic exchange. The availability of the genome sequence of *E. coli* K12 as well as of an exhaustive collection of derivative deletion mutants encouraged this presumption. However, after assaying the H47 GI excision in a number of mutants affected in recombination functions, including the 16 *E. coli* K12 site-specific recombinases, the RecET system, and multiple proteins that participate in the RecA-dependent pathways of homologous recombination, it was concluded that none of these processes appeared to be involved and that an unknown RecA-independent recombination mechanism would be responsible for the H47 GI mobility. Anyway, our results could harbor some uncertainty since the PCR technique has been described to generate chimeric products when the template contains repeated sequences, an artifact that has been detected in the amplification of 16S rRNA genes in metagenomic studies (*Wang & Wang, 1997*). Although this possibility seems rather unlikely, we cannot completely discard it.

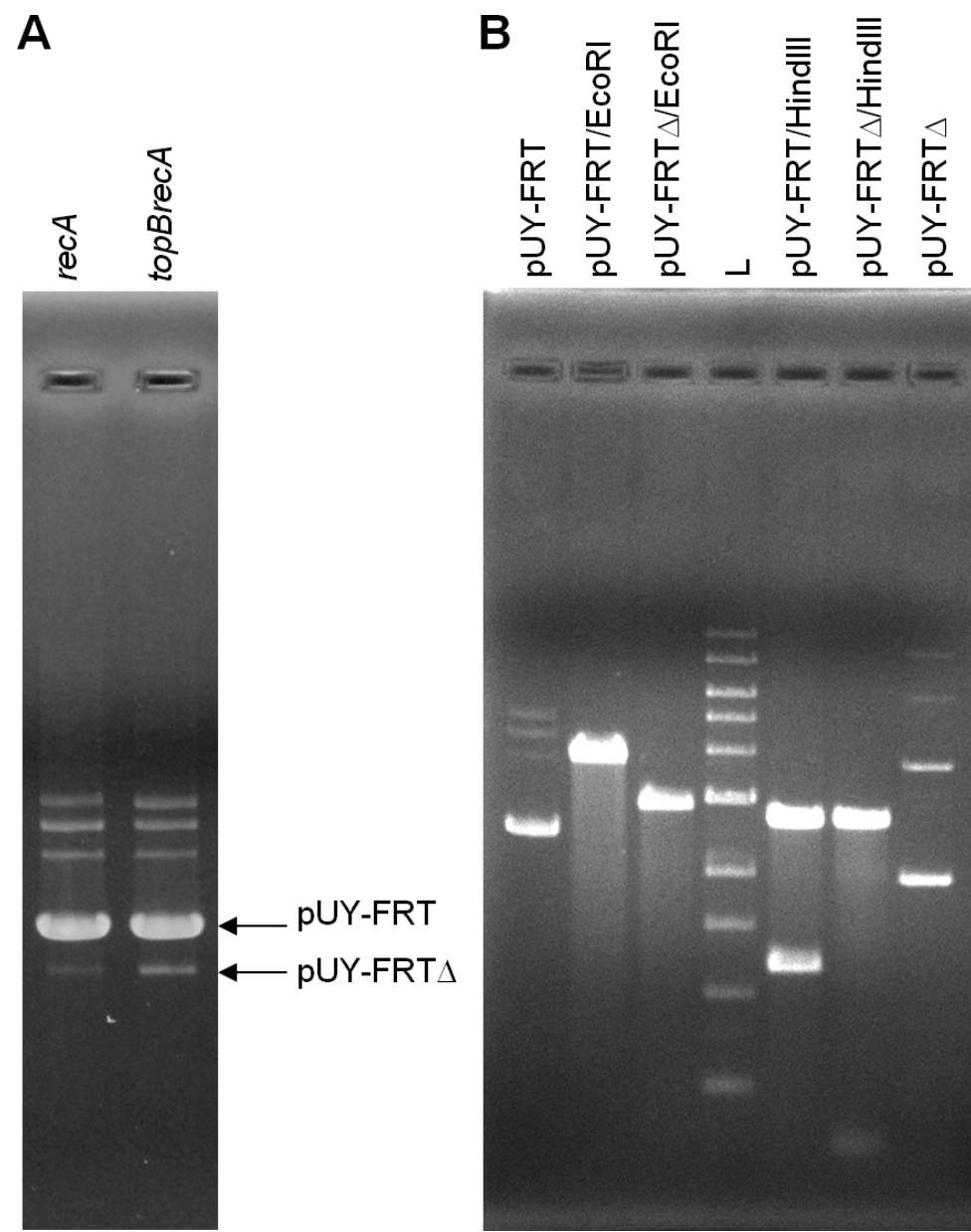

**Figure 2  Excision of the *cat* gene in pUY-FRT and generation of pUY-FRTΔ.** (A) Plasmid DNA extracted from strains BW25113 *recA* (pUY-FRT) and BW25113 *topB recA* (pUY-FRT). Bands corresponding to the original plasmid pUY-FRT (3,864 bp) and to its deletion derivative pUY-FRTΔ (2,932 bp) are indicated with arrows. (B) Restriction analysis of pUY-FRT and pUY-FRTΔ. L: 1 kb DNA Ladder (New England Biolabs, Ipswich, MA, USA).

Therefore, an *in vivo* demonstration of the existence of the H47 GI mobility was searched for. This was particularly difficult given the lack of selection for the recombined clones and that the frequency of recombination appeared to be low. In fact, we were unable to obtain a clone carrying a recombination product, i.e., an *attC* site in a replicon that had previously carried the H47 GI.

Other authors have analyzed the mobility of some genetic elements that lack a recombinase gene by means of PCR experiments similar to those employed in this work (*Palmieri, Mingoia & Varaldo, 2013*). These elements also have in common that their attachment sites are very extensive, of several hundred bp. In two cases, a defective prophage-like element from *S. enterica* serovar Enteritidis and a transposon from *Streptococcus pneumonia*, clones that spontaneously lost these elements could be isolated. Interestingly, their excision was PCR-detected in a RecA-deficient background (*Santiviago et al., 2010*; *Palmieri et al., 2012*). In view of these previous descriptions, we presume that the length of the attachment sites and their level of identity may be an important factor that determines the frequency at which recombination takes place and, thus, the likelihood of its *in vivo* detection. The two mentioned examples from *S. enterica* and *S. pneumonia* have attachment sites of 308 bp (98% of identity) and ca. 1,200 bp (>99% of identity), respectively. For their part, the H47 GI repeats have clearly smaller numbers: 146/8 bp in length with 77% of identity, the most extensive perfect homology being of 20 bp (see Fig. 1B). In other previous descriptions of RecA-independent recombination events, it was also found that homology was required and that recombination rates increased with the length of the repeats (*Albertini et al., 1982*; *Bi & Liu, 1994*). Another factor that could hinder the detection of recombined molecules was that the H47 GI encodes an antibacterial activity, microcin H47, together with it cognate immunity. Those cells harboring deleted molecules by loss of the island would be in a clear disadvantage compared to other cells because they would be more sensitive to the microcin. Nevertheless, when we worked with a plasmid construction lacking most of the island content, recombinant molecules could not be detected either.

Finally, a survey performed in data banks searching for H47 GI-related *attC* sequences revealed that these empty attachment sites exist in natural strains. Representatives of three of the four possible versions of recombination between the direct repeats were found. These findings show that the H47 GI is indeed a mobile genetic element whose presence in the bacterial chromosome is unstable. Although these empty attachment sites would represent traces of recombination events between the direct repeats, they do not ensure the widespread occurrence of this type of process in the *E. coli* species: the possibility that recombination could only take place in some genetic backgrounds providing a specific recombinase cannot be discarded.

At this point, we wondered how general this recombination phenomenon could be and then decided to study a different genetic model, plasmid pUY-FRT, in the context of *E. coli* K12. This model is much simpler than the H47 GI and contains perfect and longer direct repeats of 46 bp flanking the *cat* gene. As in the case of the island, excision events were detected by a PCR approach in wild type and in *recA-* and *recT*-deficient contexts, and the genetic exchange always occurred at the pUY-FRT repeats. However, unlike what happened with the island, this model allowed the detection of a deleted derivative molecule which had lost the *cat* gene by recombination between the direct repeats. In fact, this is probably the first time that a RecA-independent recombination product is readily detected in a plasmid preparation from a culture that had been seeded with a strain carrying the original plasmid form. This phenomenon appeared to be enhanced in a *topB* mutant, deficient

for topoisomerase III, a context that has been previously described to increase some RecA-independent recombination events (*Whoriskey, Schofield & Miller, 1991*; *Schofield et al., 1992*; *Uematsu, Eda & Yamamoto, 1997*). Therefore, in the model of pUY-FRT we succeeded to demonstrate that a RecA-independent recombination mechanism is able to work on short direct repeats in *E. coli* K12. These results support the idea that similar events could indeed happen in the H47 GI but, in this case, their frequency would be too low to be detected *in vivo*. There are several differences between the two models analyzed that could determine differences in their excision rate. For instance, it has been claimed that the longer the extent of perfect homology between the repeats and the shorter the distance separating them, the higher the rate of recombination (*Albertini et al., 1982*; *Bi & Liu, 1994*; *Lovett et al., 1993*; *Lovett et al., 1994*). These two factors could explain the higher frequency of excision in pUY-FRT in relation to that in H47 GI. pUY-FRT has 46 bp-long perfect direct repeats separated by an intervening sequence of 886 bp, while the H47 GI has perfect repeats of 13, 11, 20 and 17 bp, separated by 12,635 bp. Apparently, the condition of the DNA repeats would be predominant, considering that recombination between the H47 GI repeats in plasmid pΔint1 could not be detected *in vivo* even though the intervening sequence was shortened to 554 bp, and even in a *topB* background.

Following the results attained by PCR experiments, the deleted material in the excision events would remain as a non-replicative circle, as has been reported for other genetic elements that exhibit RecA-independent mobility (*Palmieri, Mingoia & Varaldo, 2013*). However, we could not detect the excised form *in vivo* in none of the two models analyzed. Obviously, this aspect deserves more studies.

In sum, we provide new evidence on the occurrence of spontaneous events of excision in *E. coli* K12 which are not mediated by the known mechanisms of recombination. The rearrangements under study appear to be related to previous descriptions of RecA-independent recombination events in *E. coli* K12 (*Azpiroz, Bascuas & Laviña, 2011*; *Bzymek & Lovett, 2001*; *Kingston et al., 2015*) and in other bacterial organisms (*Santiviago et al., 2010*; *Palmieri et al., 2012*). This type of genetic exchange occurs between regions of homology, including very short ones on which RecA-dependent recombination would not be able to work or would not be efficient. Since its frequency appears to be particularly low, PCR-detection of the recombination products is being routinely used by several authors, while obtaining clones carrying a recombined form has been achieved in very few cases. In this sense, RecA-independent recombination in Hfr × F⁻ crosses of extensive chromosomal segments have been reported (*Kingston et al., 2015*). Interestingly, in a recent article, these authors describe results suggesting that more than one mechanism could contribute to these RecA-independent recombination phenomena (*Kingston, Ponkratz & Raleigh, 2017*). In this work we present PCR-based evidence of RecA-independent recombination in two genetic models while *in vivo* evidence of this phenomenon was attained in only one of them. Finally, it should be kept in mind that if RecA-independent recombination were able to operate on a wide spectrum of repeated sequences, including very short ones, then it would provide mobility to different genetic elements, thus broadening the repertoire of possible rearrangements in the bacterial cell.

## ACKNOWLEDGEMENTS

We are indebted to María Parente for excellent technical assistance.

### Funding

This work was supported by the Agencia Nacional de Investigación e Innovación FCE_2_2011_1_6019 and by Programa de Desarrollo de las Ciencias Básicas, Uruguay. There was no additional external funding received for this study. The funders had no role in study design, data collection and analysis, decision to publish, or preparation of the manuscript.

### Grant Disclosures

The following grant information was disclosed by the authors:
Agencia Nacional de Investigación e Innovación.
Programa de Desarrollo de las Ciencias Básicas, Uruguay.

### Competing Interests

The authors declare there are no competing interests.

### Author Contributions

- María F. Azpiroz conceived and designed the experiments, performed the experiments, analyzed the data, contributed reagents/materials/analysis tools, wrote the paper, prepared figures and/or tables, reviewed drafts of the paper.
- Magela Laviña conceived and designed the experiments, performed the experiments, analyzed the data, contributed reagents/materials/analysis tools, wrote the paper, prepared figures and/or tables, reviewed drafts of the paper.

### Supplemental Information

Supplemental information for this article can be found online at http://dx.doi.org/10.7717/peerj.3293#supplemental-information.

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
