# Peer review of "Analysis of RecA-independent recombination events between short direct repeats related to a genomic island and to a plasmid in Escherichia coli K12"

_PeerJ, doi:10.7717/peerj.3293_

## Round 0.1 · original submission · Minor Revisions

This paper is interesting and novel. Even though Reviewer 1 has more questions about the manuscript, I believe that even if those questions should be addressed carefully, they do not represent major revision. Please address all the comments.

·

Basic reporting

The paper is well written and the context is clear. References and figures are adequate.

Experimental design

It involves original research, with a well defined question. I have a few points regarding the methods employed (see below)

Validity of the findings

I have some points regarding the conclusions (see below)

Additional comments

In this paper, the authors analyze the occurrence of RecA-independent recombination, with two different systems. With both they detect recombination events through a PCR approach, but only with one of them they are able to detect recombination products "in vivo". Also, they try to determine which site-specific or general recombinases participate in the process, but none of the enzymes checked was found to participate.
The paper is well written and the context is clear. References and figures are adequate.
It involves original research, with a well defined question. I have a few points regarding the methods employed, and the interpretation of some of the results:

1- The data reported in Figure 1 is similar to that previously reported in: Azpiroz, Bascuas & Laviña, 2011-
This should be stated.

2- The experiments to define participation of site- specific or general recombinases in the excision of H47-GI did not reveal any positive results. The authors conclude that: "it was concluded that none of these processes appeared to be involved and that an unknown RecA-independent recombination mechanism would be responsible for the H47 GI mobility" (lines 299 - 300).

One possibility is that more than one recombinase is able to perform the excision of the GI, so double mutants would be needed to identify a negative phenotype.

3- Also, the authors state that: "our results could harbor some uncertainty since the PCR technique has been described to generate chimeric products when the template contains repeated sequences, an artifact" (lines 301 - 303)

This possibility is certainly open, it would be very nice if it could be discarded in some way.

4- Related to the previous point, the authors searched for an in vivo demonstration of the existence of the H47 GI mobility, which would have helped discard the possibility of the PCR results being due to an artifact.

Unfortunately, the results were negative. I suggest that the experiments be repeated, and the deleted plasmids searched for not only directly by trying to visualize the plasmids on gels, but using hybridization, to increase the sensitivity for detection.
Another approach would be to introduce a marker that allows positive selection for loss, such as the sacR sacB genes which confer sucrose sensitivity when present, thus allowing selection for their loss by acquirement of sucrose resistance.

A third possibility would be to use one of the other E. coli or Salmonella strains which present an empty attC site, and try to isolate derivatives which have integrated the element.

5- On the other hand, using another experimental model, pUY-FRT, the authors are able to detect recombination events "in vivo", indicating that the PCR results are not an artifact.

At least for the FRT vector, even if it is suggestive, it is still possible that the H47 products are artifacts.

Reviewer 2 ·

Basic reporting

No comment

Experimental design

No comment

Validity of the findings

No comment

Additional comments

Azpiroz and Laviña further investigate the mobility of an E. coli K12 genomic island (H47) that they had previously shown to be excised in circular form from the chromosome, thanks to direct repeats, in the absence of recombinase genes. Since, in that previous study, the mobility of H47 had been assumed to depend on trans-acting factors provided by E. coli K12, the point is now to understand which host factors are actually involved. Although the authors don’t succeed to conclusively disclose the underlying mechanism, they provide interesting new evidence that RecA-independent recombination may be involved.

The authors cope well with the uneasy topic, and both experimental work and arguments are adequate. Only a few minor comments:

1. In the title, rather than generically speaking of ‘a genomic island’, the authors could directly speak of genomic island H47.

2. Readers would probably appreciate a clearer explanation of what the authors mean by ‘in vivo confirmation’ experiments.

3. In both Results (lines 208-210) and Discussion (lines 331-332), the authors say they investigated the H47 GI excision also in a recombinant plasmid in which most of the sequence between the att sites was deleted. Did they check whether excision also occurred in recombinant plasmids where either att site was deleted?

4. The whole manuscript would take advantage of some shortening and streamlining.

Reviewer 3 ·

Basic reporting

Could use a few more recent references on the topic.

Experimental design

No Comment

Validity of the findings

No comment

Additional comments

This paper builds on a previous study examining excision of a genomic island in E. coli. The GI does not contain its own integrase but does contain att sites suggesting site-specific recombination may play a role. In order to determine whether the island hijacks an integrase function elsewhere on the chromosome, they examined a set of recombinase mutants. However, they did not identify a specific recombinase, which lead them to examine RecA dependent recombination. These studies also did not lead to a defined mechanism for the excisive recombination.
This is a clear well written and well executed study that unfortunately did not lead to an answer to the original questions, which is likely a very difficult task due to possible redundancies in recombinase function among some integrases. The information gather by the authors should be very useful to those in the field.

---

## Round 0.2 · accepted · Accept

This issues that concerned the reviewers have been addressed properly.